# Aza-Reversine Promotes Reprogramming of Lung (MRC-5) and Differentiation of Mesenchymal Cells into Osteoblasts

**DOI:** 10.3390/ma14185385

**Published:** 2021-09-17

**Authors:** Fani Tsitouroudi, Vasiliki Sarli, Dimitrios Poulcharidis, Maria Pitou, Alexandros Katranidis, Theodora Choli-Papadopoulou

**Affiliations:** 1Department of Chemistry, Aristotle University of Thessaloniki, University Campus, 54124 Thessaloniki, Greece; fanioli@gmail.com (F.T.); dimitris.poulcharidis@hotmail.com (D.P.); margeopit@chem.auth.gr (M.P.); 2Institute of Biological Information Processing IBI-6, Forschungszentrum Jülich (FZJ), 52425 Jülich, Germany; a.katranidis@fz-juelich.de

**Keywords:** aza-reversine, cell reprogramming, MRC-5 cells, mesenchymal cells, immobilization of aza-reversine, PMMA-HEMA surfaces

## Abstract

Reversine or 2-(4-morpholinoanilino)-N6-cyclohexyladenine was originally identified as a small organic molecule that induces dedifferentiation of lineage-committed mouse myoblasts, C2C12, and redirects them into lipocytes or osteoblasts under lineage-specific conditions (LISCs). Further, it was proven that this small molecule can induce cell cycle arrest and apoptosis and thus selectively lead cancer cells to cell death. Further studies demonstrated that reversine, and more specifically the C2 position of the purine ring, can tolerate a wide range of substitutions without activity loss. In this study, a piperazine analog of reversine, also known as aza-reversine, and a biotinylated derivative of aza-reversine were synthesized, and their potential medical applications were investigated by transforming the endoderm originates fetal lung cells (MRC-5) into the mesoderm originated osteoblasts and by differentiating mesenchymal cells into osteoblasts. Moreover, the reprogramming capacity of aza-reversine and biotinylated aza-reversine was investigated against MRC-5 cells and mesenchymal cells after the immobilization on PMMA/HEMA polymeric surfaces. The results showed that both aza-reversine and the biofunctionalized, biotinylated analog induced the reprogramming of MRC-5 cells to a more primitive, pluripotent state and can further transform them into osteoblasts under osteogenic culture conditions. These molecules also induced the differentiation of dental and adipose mesenchymal cells to osteoblasts. Thus, the possibility to load a small molecule with useful “information” for delivering that into specific cell targets opens new therapeutic personalized applications.

## 1. Introduction

Stem cells are unique cells that are able to self-renew and differentiate into specific cell types under proper conditions. Their key role in cellular processes during embryonic development and tissue homeostasis stem cells offer great potential in the treatment of various diseases, including type I diabetes [1,2], muscular dystrophies, bone diseases, cancer, neurodegenerative and cardiovascular [3] disorders [4]. Major drawbacks in their applications arise from their scarce availability and difficulty in controlling cell fate. However, recent achievements have allowed the reprogramming of somatic cells to induced pluripotent stem cells by defined genetic factors or by small molecules [5]. In this context, reversine was identified as a novel dedifferentiation factor of lineage-committed mouse myoblasts C2C12, which were redirected into lipocytes or osteoblasts, under lineage-specific conditions (LISCs) [6]. Since then, reversine has become a particularly useful tool for the study of the control of cell fate. According to Anastasia and coworkers, reversine was able to transform primary murine and human dermal fibroblasts into myogenic-competent cells in vitro and in vivo [7]. It also induced adipocyte differentiation in 3T3-L1 cells [8] and increased the plasticity of C2C12 myoblasts towards the neuroectodermal line [9]. Lineage-committed annulus fibrosus cells dedifferentiated into progenitor-like cells after treatment with reversine [10] and porcine muscle-derived stem cells (PMDSCs) were transformed into female germ-like cells based on the reports of Saraiya and Lv, respectively [11]. All the aforementioned studies focused on the effect of reversine on mesoderm-derived cells, which were transformed into cells of the mesodermal, endodermal or germ line. Further studies revealed that reversine can be employed not only as a dedifferentiation factor but also for the enhancement of the differentiation ability of mesenchymal stem cells [12].

Considering the molecular targets of reversine, it was previously demonstrated that it is a moderately potent antagonist for the human A3 adenosine receptor with a Ki value of 0.66 μM [13] and a dual inhibitor of MEK1 (IC50 > 1.5 μΜ) and nonmuscle myosin II heavy chain (IC50 > 0.35 μΜ) [14]. Moreover, it inhibits multiple mitotic kinases, such as Aurora kinase A (IC50 = 0.15 μΜ), B (IC50 = 0.5 μΜ), C (IC50 = 0.4 μΜ) and human MPS1 with an IC50 value of 2.8 nΜ [15,16,17]. Further preliminary studies on reversines’ analogs that exhibit the same or greater effects have been conducted. A structure-activity relationship (SAR) analysis gave light to reversine’s substitution sites and concluded that its activity is determined by both the N9 hydrogen and the NH substitution at the C2 position of the purine ring. On the contrary, dedifferentiating effects are retained after substitution of primary amines at the C6 position of the purine ring [18].

In this context, and along with the fact that there are no published data for reversine immobilization, a reversine analog that can be tagged with biotin was synthesized. The biotin-streptavidin bond compromises the strongest non-covalent bond occurring naturally and is used widely for the immobilization of molecules on materials or nanotubes in applications of nanotechnology. Thus, the biotinylation of reversine allows its immobilization on nanomaterials for manufacturing transplants or on nanocarriers for selective cancer treatment. This study was orientated towards the synthesis of a reversine analog, aza-reversine, that can be biotinylated via a peptide bond formation, the effect of this analog and its biotinylated counterpart on MRC-5 cell reprogramming to osteoblasts under osteogenic-inducing conditions. Further, their differentiation capacity on mesenchymal stem cells was also evaluated and discussed in this paper.

## 2. Methods

### 2.1. General Experimental Details

All reactions were carried out under an atmosphere of Ar unless otherwise specified. All commercial reagents were used without further purification. Reactions were monitored by TLC with visualization by UV light and visualizing agent aqueous ceric sulfate/phosphomolybdic acid or ethanolic p-anisaldehyde solution. ^1^H and ^13^C NMR spectra were acquired on NMR instruments (Bruker Avance III 300, Madison, WI, USA; Agilent 500 MHz, Palo Alto, CA, USA) at 300, 500 and 75, 125 MHz, respectively. Mass spectra data were acquired on a mass spectrometer (Hybrid Ion Trap-Orbitrap Mass Spectrometer, Thermo Fisher Scientific, Waltham, MA, USA) equipped with an electrospray ion source in positive mode (source voltage 3.5 kV, sheath gas flow 10, capillary temperature 275 °C) with resolution R = 60.000 at *m*/*z* = 400 (mass range = 150–2000) and dioctylphtalate (*m*/*z* = 391.28428) as the “lock mass”. The surface of the samples was analyzed by infrared spectroscopy using an FTIR spectrometer Spectrum One (Perkin Elmer, Wellesley, Boston, MA, USA) and was analyzed in triplicates. A horizontal attenuated total reflection (HATR) accessory was used with a Zinc Selenide (ZnSe) plate, and each spectrum was obtained between 4000 and 600 cm^−1^ with a resolution of 4 cm^−1^. Data processing and calculation were conducted with the commercial software Spectrum v5.0.1 (Perkin Elmer LLC 1500F2429, Waltham, MA, USA).

### 2.2. Tert-Butyl 4-(4-Aminophenyl)piperazine-1-carboxylate, S1

To a solution of 1-(4-aminophenyl)piperazine (350 mg, 1.97 mmol) in anhydrous CH_2_Cl_2_ (11 mL) at 0 °C, di-*tert*-butyl di-carbonate (406 mg, 1.86 mmol) was added. The resulting mixture was stirred at 0 °C for 2 h, poured into brine (10 mL), alkanilized with 1N NaOH and extracted with CH_2_Cl_2_ (3 × 15 mL). The combined organic layers were washed with H_2_O, dried (Na_2_SO_4_) and the solvent was evaporated under vacuum. Purification by flash chromatography (eluent; hexane/ethyl acetate = 1/1) afforded S1 (413 mg) in 80% yield as a brown solid. ^1^H NMR data is in accordance with the literature [19]. S1: ^1^H NMR (500 MHz, CDCl_3_) δ 6.80 (d, *J* = 8.5 Hz, 2H), 6.65 (d, *J* = 8.5 Hz, 2H), 3.56 (s, 4H), 2.96 (s, 4H), 1.47 (s, 9H); ^13^C NMR (125 MHz, CDCl_3_) 154.7, 144.4, 140.6, 119.2, 116.1, 79.7, 51.2, 28.4; HRMS *m*/*z* for C_15_H_24_N_3_O_2_ [M+H]^+^ calculated 278.1869, found 278.1860 (Appendix A).

### 2.3. N-Cyclohexyl-2-fluoro-9H-purin-6-amine, S2

Cyclohexylamine (90 μL, 0.782 mmol) and DIPEA (181 μL, 1.04 mmol) were added successively to a solution of compound 3 (150 mg, 0.869 mmol) in *n*-butanol (3 mL). Subsequently, the reaction mixture was heated to 80 °C with vigorous stirring for 24 h. The solvent was evaporated, and the crude material was used for the next step without further purification. HRMS *m*/*z* for C_11_H_15_FN_5_ [M+H]^+^ calcd 236.1311, found 236.1306 [13].

### 2.4. Tert-Butyl 4-(4-(6-(Cyclohexylamino)-9H-purin-2-ylamino) phenyl) piperazine-1-carboxylate, 4

A mixture of 2-fluoro-6-cyclohexylamino-purine (0.869 mmol) in ethanol (1.7 mL) and S1 (482 mg, 1.738 mmol) was stirred at 110 °C for 48 h. The mixture was evaporated in vacuum and the crude material was purified by flash chromatography (eluent; ethyl acetate/methanol/triethylamine = 10/0.5/0.02) to afford 346 mg of 4 as a yellow solid (81% yield over two steps). 4: m.p. = 96–98 °C ; ^1^H NMR (500 MHz, CDCl_3_) δ 12.78 (s, 1H), 7.43 (d, *J* = 8.3 Hz, 2H), 6.91 (d, *J* = 8.3 Hz, 2H),6.73 (s, 1H), 5.57 (s, 1H), 4.08 (s, 1H), 3.58 (s, 4H), 3.07 (s, 4H), 2.09 (d, *J* = 12.4 Hz, 2H), 1.78 (d, *J* = 12.7 Hz, 2H), 1.66 (d, *J* = 12.4 Hz, 1H), 1.49 (s, 9H), 1.42 (dd, *J* = 24.7, 12.6 Hz, 1H), 1.35—1.15 (m, 4H); ^13^C NMR (126 MHz, CDCl_3_) δ 157.1, 154.7, 154.4, 147.8, 135.6, 132.9, 123.0, 117.8, 117.3, 114.5, 79.8, 50.2, 49.2, 43.5, 33.3, 28.4, 25.6, 24.9; FT-IR: 3472, 2873, 1652, 1558, 1540, 1496, 1457, 1305, 1275, 1195, 1016, 958, 871; HRMS *m*/*z* for C_26_H_37_N_8_O_2_ [M+H]^+^ calcd 493.3039, found 493.3016 (Appendix A).

### 2.5. N6-Cyclohexyl-N2-(4-(piperazin-1-yl) phenyl)-9H-purine-2,6-diamine, 2

Compound 4 (93 mg, 0.189 mmol) was dissolved in 1 mL 5N HCl in isopropanol and the mixture was stirred at room temperature for 2 h. The solution was then neutralized with NaHCO_3_ and the aqueous layer was extracted with ethyl acetate (3 × 3 mL). The combined extracts were dried (Na_2_SO_4_), filtered and concentrated. The crude material was purified by flash chromatography (eluent; ethyl acetate/methanol/ammonium hydroxide = 10/1/0.01) to afford 72 mg of 2 in a 97% yield. 2: 1H NMR (500 MHz, CD_3_OD) δ 7.73 (s, 1H), 7.58 (d, J = 8.5 Hz, 2H), 6.93 (d, J = 8.7 Hz, 2H), 4.09 (s, 1H), 3.07 (dd, J = 19.0, 5.1 Hz, 4H), 2.98 (d, J = 3.8 Hz, 4H), 2.10 (d, J = 12.2 Hz, 2H), 1.83 (d, J = 13.3 Hz, 2H), 1.69 (d, J = 12.6 Hz, 1H), 1.59–1.13 (m, 5H); ^13^C NMR (126 MHz, CD_3_OD) δ 158.9, 155.1, 152.8, 148.1, 137.6, 136.3, 121.8, 118.7, 114.1, 52.7, 50.6, 46.7, 34.2, 27.0, 26.3; FT-IR: 3413, 2927, 2851, 1617, 1512, 1478, 1414, 1240, 1151, 1043, 925, 821; HRMS *m*/*z* for C_21_H_29_N_8_ [M+H]^+^ calcd 393.2515, found 393.2508 (Appendix A).

### 2.6. (3. aS,4S,6aR)-4-(5-(4-(4-((6-(Cyclohexylamino)-9H-purin-2-yl)amino)phenyl)piperazin-1-yl)-5-oxopentyl) Tetrahydro-1H-thieno[3,4-d]imidazol-2(3H)-one, 5

To a solution of compound 2 (25 mg, 0.064 mmol) in 0.6 mL dry DMF 1-ethyl-3-(3-dimethylaminopropyl)carbodiimide (13 mg, 0.067 mmol), N,N-dimethylaminopyridine (10 mg, 0.081 mmol) and 15 mg (0.063 mmol) of biotin at 0 °C were added. Next, the reaction mixture was stirred overnight at the ambient temperature. The solution was then concentrated under vacuum and the residue was purified by flash chromatography (eluent; dichloromethane/methanol/NH_4_OH = 7/1/0.01) to afford 30 mg of 5 as a white solid (75% yield). 5: m.p.= 223–225 °C; ^1^H NMR (500 MHz, CDCl_3_-CD_3_OD) δ 7.57–7.50 (m, 3H), 6.92 (d, J = 8.6 Hz, 2H), 4.60–4.42 (m, 1H), 4.41–4.27 (m, 1H), 4.06 (s, 1H), 3.78 (brs, 4H), 3.65 (s, 4H), 3.39 (s, 1H), 3.12 (m, 4H), 2.83 (1H missing due to overlapping), 2.72 (d, J = 12.9 Hz, 1H), 2.41 (t, J = 7.2 Hz, 2H), 2.11 (d, J = 10.9 Hz, 2H), 1.83–1.67 (m, 7H), 1.58–1.20 (m, 7H); ^13^C NMR (126 MHz, CDCl_3_-CD_3_OD) δ 171.6, 163.7, 156.8, 153.9, 146.1, 135.1, 134.1, 120.9, 120.8, 117.7, 113.6, 61.7, 59.9, 55.3, 50.4, 45.6, 41.6, 40.3, 32.8, 32.5, 29.5, 28.4, 28.1, 25.5, 24.8; FT-IR: 3276, 2924, 2853, 1685, 1624, 1560, 1508, 1425, 1327, 1226, 1150, 1117, 1031, 979, 945, 823, 767; HRMS *m*/*z* for C_31_H_43_N_10_O_2_S [M+H]^+^ calcd 619.3291, found 619.3282 (Appendix A).

### 2.7. Chemical Modification of Polymeric PMMA/HEMA Surfaces

According to previous studies, PMMA/HEMA surfaces can be further modified to biofunctionalize a great variety of molecules. Compound 5 was immobilized on PMMA/HEMA surfaces through the non-covalent bond of streptavidin-biotin, as previously described [20]. It is known that streptavidin can simultaneously bind four biotin molecules. Thus, PMMA/HEMA surfaces were modified with biotin groups in order to non-covalently bind streptavidin and laterally bind the biotinylated molecule.

### 2.8. Cell Culture

Dental pulp mesenchymal stem cells were kindly provided by Assistant Professor A. Bakopoulou from the School of Dentistry, Aristotle University of Thessaloniki, and were collected by the enzymatic dissociation method described in Bakopoulou et al. (2016) [21]. The samples had been collected in accordance with all the relevant guidelines and regulations and had been approved by the Institutional Review Board of the Aristotle University of Thessaloniki (Nr. 66/18 June 2018). Adipose mesenchymal cells were kindly provided by Professor G. Koliakos from Biohellenika A.E. Lung cells (MRC-5) were supplied by AΤCC (CCL-171). Mesenchymal cells and MRC-5 cells were cultured in Dulbecco’s modified eagle medium (Gibco) supplemented with 10% fetal bovine serum (Gibco) and 1% penicillin/streptomycin mixture (Gibco) at 37 °C in a humidified atmosphere containing 5% CO_2_. After reaching 80% of confluency, cells were collected with PBS 1Xusing a spatula and were centrifuged at 1100 rpm for 3 min. The cell pellet was resuspended in a complete medium (DMEM supplemented with 10% FBS and 1% penicillin/streptomycin). MRC-5 cells and mesenchymal stem cells within the early passage (passage 2) were placed in 12-well plates in 5000 cells/well and 20,000/well, respectively. After 24 h, attached cells were treated with aza-reversine or its biotinylated form (100 nΜ/well) for 24 h for mesenchymal cells or 96 h for MRC-5 cells. After the elapsed time, the medium of mesenchymal or MRC-5 cells was replaced with osteogenic medium (StemPro^®^ Osteogenesis Differentiation Kit, Invitrogen, Waltham, MA, USA), respectively. Every 2–3, days the osteogenic medium was renewed.

Regarding cell cultures on modified surfaces with aza-reversine 5, they were placed in 12-well plates and cells were added to them in an appropriate number (5000 MRC-5 cells/well, 20,000 mesenchymal stem cells/well). The cells were cultured with osteogenic medium surfaces, which were changed every 2–3 days.

### 2.9. Alkaline Phosphatase Assay

Alkaline phosphatase consists of a remarkable biomarker used during in vitro bone formation and osteogenic differentiation, and its activity is enhanced during these processes. It can be detected in proliferating osteoblasts during the 7th and 14th day of osteogenesis, which are stained blue-purple, while control cells are not painted (control mesenchymal cells may differentiate spontaneously in a small percentage). After 14 days of differentiation, the osteogenic medium was removed from the wells, and the cells were washed with PBS 1X. Cells were treated with neutral buffered formalin (10% *v*/*v*) for 60 s, and after the elapsed time, the neutral buffered formalin was removed. The cells were washed with washing buffer (0.05% *v*/*v* Tween-20 to Dulbecco’s PBS), and the substrate solution BCIP/NBT was added to the cultures. After incubation for 5–10 min at RT in the dark, the substrate solution was removed, cells were washed with washing buffer, and PBS 1X was added in the end. Images were taken with a Nikon Microscope at 40× magnification.

### 2.10. Alizarin Red Staining

Alizarin red staining can detect extracellular calcium deposits formed during in vitro osteogenesis. By using this assay, differentiated cells appear bright orange-red while undifferentiated cells are slightly reddish-pink or colorless. After 21 days of differentiation, the osteogenic medium was removed, cells were washed with Dulbecco’s PBS (Gibco) and treated with neutral buffered formalin (10% *v*/*v*) for 30 min. Afterward, neutral buffered formalin was removed, cells were washed with distilled water and treated with alizarin red staining solution (2% *w*/*v* in distilled water, pH 4.1–4.3) for 20 min at RT in the dark. After the elapsed time, the staining solution was removed, cells were washed four times with 1 mL distilled water, and PBS was added at the end. Images were taken with a Nikon Microscope at 40× magnification.

### 2.11. Reverse-Transcription Polymerase Chain Reaction (rt-PCR)

After 12 days of osteogenic differentiation, cells were collected with PBS 1x using a spatula. Total RNA was isolated from cells with a NucleoSpin^®^ RNA/Protein kit (Macherey-Nagel), and rt-PCR was performed for osteocalcin (OC), which is a specific marker of osteoblasts and for b-glucuronidase (GusB), which is a housekeeping gene and was used as control. The primers that were used for osteocalcin were: forward primer 5′-ACACTCCTCGCCCTATTG-3′ and reverse primer 5′-GATGTGGTCAGCCAACTC-3′. The primers used for b-glucuronidase were: forward primer 5′-TACGAACGGGAGGTGATCCT-3′ and reverse primer 5′-TGGCGATAGTGATTCGGAGC-3′. The rt-PCR reaction was conducted with a RobusT™I kit (Finnzymes) using the following conditions: RNA denaturation at 68 °C for 2 min, cDNA synthesis at 50 °C for 10 min and at 55 °C for another 10 min, deactivation of AMV-RT and denaturation of mRNA-cDNA hybrid at 94 °C for 2 min. cDNA was amplified for 30 cycles with the following conditions: denaturation at 94 °C for 1 min, annealing at 60 °C for 1 min, extension at 72 °C for 1 min followed by 5 min at 72 °C at the end of 30 cycles. The samples were electrophoresed on agarose gel at 100 Volt for 25 min.

## 3. Results and Discussion

### 3.1. Chemical Synthesis of Aza-Reversine 2 and Biotinylated Aza-Reversine 5

Previous structure-activity relationship studies by Schultz and coworkers [14] showed that the C2 position of the purine ring in reversine can tolerate a wide range of substitutions. In consequence, aza-reversine 2 was chosen as a suitable substrate because it can be conjugated with a biotin tag via a peptide bond formation with the free amino group in the piperazine ring (Figure 1). Substitution of the commercially available 6-chloro-2-fluoropurine (3) first with cyclohexylamine following the protocols reported by Ding [6] and Perreira [13] and then with tert-butyl 4-(4-aminophenyl)piperazine-1-carboxylate afforded the key intermediate 4 at an 81% yield (over the two steps) [19]. After removal of the N-Boc group in 4 using 4N HCl in isopropanol, reversine (aza-reversien 2) was obtained at a 97% yield. Finally, as outlined in Figure 1, biotin was coupled to aza-reversine 2 by using ECDI in DMF with DMAP at a 75% yield.

### 3.2. Aza-Reversine 2 Induces Reprogramming of MRC-5 Cells to Osteoblasts and Differentiation of Mesenchymal Cells to Osteoblasts

Aza-reversine 2 was tested for its ability to induce the reprogramming of lung cells (MRC-5). MRC-5 cells were treated with 100 nM of aza-revesrine 2 (diluted in <1% DMSO) for 96 h, while control cells were treated only with DMSO. As shown in Figure 1, control cells increased in number and maintained their original form, while treated cells stopped proliferating and developed long branches, in accordance with previous studies [17]. MRC-5 cells treated with aza-reversine 2 seem to be multinuclear (polyploidy). This phenotype is attributed to the inhibition of MPS1 kinase that is essential for the spindle assembly checkpoint [17].

After 96 h, the medium was removed and replaced with an osteogenic differentiation medium (StemPro^®^ Osteogenesis Kit, Invitrogen), and cells were collected after 12, 14 and 21 days of differentiation. On the 12th day of differentiation, mRNA was isolated from MRC-5 cells for testing the expression of osteocalcin with rt-PCR. Osteocalcin is a marker of osteogenesis and indicates the successful differentiation of cells into osteoblasts [22]. Gus-b (Beta-glucuronidase) was chosen as a housekeeping gene, and the results showed that treated cells expressed osteocalcin at a higher degree than untreated cells. Similar results were obtained with alkaline phosphatase assay and alizarin red stain. After 14 days, cells were tested for alkaline phosphatase activity, a characteristic marker of proliferating osteoblasts. Differentiated cells were colored blue-purple, while control cells were not stained [23,24]. Finally, after 21 days of osteogenic differentiation, treated cells were stained with alizarin red, a marker of extracellular calcium deposits formed during in vitro differentiation of osteoblasts. The results showed that cells treated with aza-reversine had a higher concentration of calcium deposits compared to control cells. Overall, the high expression of osteocalcin and the positive staining with alizarin red and alkaline phosphatase indicated that the cells treated with aza-reversine 2 were reprogrammed successfully and turned to osteoblasts under LSICs (lineage-specific inducing conditions) (Figure 2) and proposed that this modification did not alter the activity of reversine.

As mentioned above, a significant property of reversine is the ability to differentiate dental and adipose mesenchymal cells into osteoblasts. In another experiment, mesenchymal cells derived from dental tissue and adipose tissue were also treated with aza-reversine 2, as previously described, in order to investigate if this property is affected after substitution. Osteogenic differentiation was tested with alizarin red stain and alkaline phosphatase activity. As shown in Figure 3, mesenchymal cells differentiated into osteoblasts after treatment with aza-reversine 2 at a concentration of 100 nM for 48 h and cultured under osteogenic inducing conditions. These results are similar to previous findings, which show that reversine enhances the differentiation ability of mesenchymal cells [12].

### 3.3. Biotinylated Aza-Reversine 5 Reprograms MRC-5 Cells Prior and after Immobilization on Polymeric Surfaces

Biotinylated aza-reversine 5 was also tested for its reprogramming and differentiation capacity on MRC-5 cells (Figure 4). MRC-5 cells were cultured and treated with biotinylated aza-reversine 5, similar to aza-reversine 2 as mentioned in paragraph 3.2. After reprogramming, cells were cultured with an osteogenic differentiation medium and were tested for alkaline phosphatase activity and alizarin red stain (on the 14th and 21st day of differentiation, respectively). As shown in Figure 4, treated cells were stained positively in both assays demonstrating that biotinylation did not affect the primitive properties of aza-reversine.

After confirming that derivatives 2 and 5 are able to reverse the lineage-committed state of MRC-5 cells to a more pluripotent state, further studies were conducted in order to investigate the differentiation capacity of immobilized aza-reversine 5 on polymeric surfaces. A variety of osteogenesis-inducing molecules immobilized on polymeric surfaces has been previously reported and applied to the design of implants used for the reconstruction of injured bone tissue [25,26,27]. According to our previous results, PMMA/HEMA polymeric surfaces are of great interest because they are non-toxic and can be easily biofunctionized with macromolecules [20]. Thus, it was interesting to examine the activity of polymeric surfaces carrying biotinylated aza-reversine 5 for further use in biological and medical applications. After the synthesis of the PMMA/HEMA polymeric surfaces as described in our previous study, the immobilization of aza-reversine on PMMA/HEMA polymeric surfaces was achieved through the non-covalent bond of streptavidin-biotin and verified by FTIR spectroscopy (See Appendix A). After the immobilization of biotinylated aza-reversine 5 on PMMA/HEMA surfaces, MRC-5 cells were cultured on them under osteogenic inducing conditions as described in Section 3.1 (Figure 5).

As recorded in the literature, biotinylated reversine shows similar effects on the differentiation of mesenchymal [1]. In our study, biotinylated aza-reversine 5 had similar reprogramming effects as aza-reversine 2 on mesenchymal cells, as shown in Figure 6.

Extracted alizarin red staining was quantified, and the results obtained showed that immobilized and free aza-reversine 2 seems to have a quite similar effect on MRC-5 cells, while on mesenchymal cells, free aza-reversine induced the formation of calcium deposits at a higher degree.

In agreement with the previous results, immobilized aza-reversine remained active and induced MRC-5 cell reprogramming and mesenchymal cell osteogenic differentiation in a similar way to reversine. Alkaline phosphatase activity and alizarin red staining exhibited the differentiating capacity of immobilized aza-reversine 5 on PMMA/HEMA surfaces. As depicted in Figure 7, in MRC-5 cells, the high osteogenic potential is obtained with immobilized aza-reversine while in mesenchymal stem cells, free aza-reversine leads to high differentiation.

## 4. Conclusions

The current study focuses on the dual role of reversine as an agent that promotes cell dedifferentiation to a primitive state and decreases tumor progression. These key roles and the subsequent substitution without activity loss make reversine a promising molecule in a variety of medicinal applications. On this basis, a reversine analog, aza-reversine 2, and its biotinylated derivative, 5, were successfully synthesized and tested for their ability to reprogram MRC-5 cells under osteogenic inducing conditions. Further, these derivatives were tested for their ability to differentiate mesenchymal stem cells into osteoblasts. Both aza-reversine and its biotinylated derivative induced in vitro osteogenesis of MRC-5 cells and mesenchymal cells under LISCs at a concentration of 100 nM. The concurrent conservation of activity after immobilization of the biotinylated derivative on polymeric surfaces through the strong non-covalent bond of streptavidin-biotin opens new horizons in biofunctionalized material construction that have both antitumorigenic and osteogenic capacities. The treatment of cancer cells co-includes several factors that have to be attacked by specific drugs. The usual problem that has to be faced nowadays is that the differences between cancer and non-cancer cells are not successfully distinguished. In particular, the specific delivery of the drug to its target requires cutting-edge, innovative technology with “intelligent substrates” capable of spontaneous target recognition as well as drug delivery. Thus, nanoparticles can be functionalized with biotinylated aza-reversine acting as an anti-cancer agent along with another molecule that recognizes healthy from diseased cells (cancer cells). The above suggestion is strongly based on the properties of this derivative regarding its capacity to be immobilized on thin layers, nanoparticles and scaffolds without abolishing its main function, which in low concentrations is the reprogramming of cells and in higher concentrations the killing agent of cancer cells. Further studies to determine the antitumorigenic capacity of these analogs in free and in biofunctionalized forms are needed.

## Data Availability

Not applicable.

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
