# Peer review of "Aza-Reversine Promotes Reprogramming of Lung (MRC-5) and Differentiation of Mesenchymal Cells into Osteoblasts"

_materials, 2021, doi:10.3390/ma14185385_

Round 1

Reviewer 1 Report

The authors have tested functions of a new analogue of reversine in reprogramming and differentiation. The text needs to be modified to be more structured for easier understanding. 

  1. The abstract needs to be rewritten with a clear rationale for this project, key experiments performed, observations made and most important findings from this study. 
  2. The introduction needs to include  a section on why authors wanted to modify reversine. What are the advantages of creating a reversine analogue and why was it necessary.
  3.  In the results section, authors need not explain methods in detail or mention source of each reagent (line 237-246). 
  4. The images presented in figure 1 are low quality and need to be replaced by higher resolution images. Also include scale bars in images.
  5. Figure 2 images are low resolution too and need to be replaced with better images. rt-PCR results are not explained in figure 2. Moreover the data from rt-PCR is not clear and quality needs to be updated. Authors can quantify data from rt-PCR and show data as graphs instead of blots. 
  6. Figures 5 and 6 needs to be replaced with better quality images. Right not, resolution is too low and I cannot interpret anything. 
  7. Text in results section needs to be rewritten to indicate why an experiment is being done, followed by clear explanation of results obtained and then interpretations from that experiment. The authors have to follow this for every section in the results.
  8. Expand conclusions section to discuss your findings. What are the implications of your study? How does this add to existing literature on this topic? 

Author Response

Aristotle University of Thessaloniki

Department of Chemistry

Lab of Biochemistry

Aristotle University of Thessaloniki

                                        54124   Thessaloniki, Greece

Professor

Theodora Choli-Papadopoulou

Tel: ++302310997806

Fax: ++302310997689

Editor-In-Chief,

Bone Reports

Thessaloniki, 10/09/2021

Dear Editors,

We would like to thank you for the opportunity to resubmit a revised copy of our manuscript entitled " Aza-reversine Promotes Reprogramming of Lung (MRC-5) and Differentiation of Mesenchymal Cells into Osteoblasts". We would also like to thank the Reviewers for the effort and time they put in revising this manuscript and for the constructive comments for correction and improvement. We have made every attempt to fully address these comments and we believe these revisions have resulted in a significantly improved manuscript.

Major modifications were made in order to discuss and explain the most important findings as clearly as possible. In the following point-by-point response, each comment is answered individually and the page number that contains the relevant information is indicated in brackets.

Responses to reviewers’ comments

First of all, we would like to thank the Reviewers for their effort and the helpful comments for improvement of the manuscript. A point-by-point response to each comment follows.

Reviewer 1

  1. “The abstract needs to be rewritten with a clear rationale for this project, key experiments performed, observations made and most important findings from this study”

The Abstract has been modified significantly to explain the most important findings more clearly (page 1).

  1. The introduction needs to include a section on why authors wanted to modify reversine. What are the advantages of creating a reversine analogue and why was it necessary”

This section has been added in the Introduction (page 1).

  1. “ In the results section, authors need not explain methods in detail or mention source of each reagent (line 237-246)”

The results section has been corrected (pages 3-7).

  1. “The images presented in figure 1 are low quality and need to be replaced by higher resolution images. Also include scale bars in images”

The images have been replaced and scale bars are included (page 4).

  1. “Figure 2 images are low resolution too and need to be replaced with better images. rt-PCR results are not explained in figure 2. Moreover the data from rt-PCR is not clear and quality needs to be updated”

The images have been replaced. Quality of rt-PCR results has been ameliorated. Also, quantification of band intensity with GelAnalyser has been added in the supplementary data (page 5).

  1. “Figures 5 and 6 needs to be replaced with better quality images. Right not, resolution is too low and I cannot interpret anything”

The images have been replaced (pages 6 and 7).

  1. Text in results section needs to be rewritten to indicate why an experiment is being done, followed by clear explanation of results obtained and then interpretations from that experiment. The authors have to follow this for every section in the results”

The results section has been rewritten (pages 3-7).

  1. “Expand conclusions section to discuss your findings. What are the implications of your study? How does this add to existing literature on this topic? “

Conclusions have been expanded and the implications and main findings have been summarized (page 7)

Reviewer 2

  1. “Reversine was originally identified as a small molecule that induces dedifferentiation to C2C12 myoblasts, which were redirected into lipocytes or osteoblasts under lineage specific conditions (LISCs)” – dedifferentiation to C2C12 from which cells? For which clinical needs would that be relevant for? Then you redirect them into lipocytes or osteoblasts? Were you trying to show differentiation potential, or something else? Indeed, this first part of your abstract is a bit unclear. Please justify these aspects a little further.

The Abstract has been modified significantly in order to explain the most important findings and the aforementioned aspects more clearly (page 1)

  1. “After aza-reversine 2 synthesis” – after synthesis of a piperazine analogue of reversine, also known as aza-reversine 2

This sentence has been replaced (page 1).

  1. Could you then focus on a particular application, such as cancer treatment, as you indicate in the Introduction section? “

Within the text of the revised manuscript with included a detailed suggestion regarding a possible application of the derivative biotinylated-aza-reversine (see page 7 at the conclusion section).

  1. “Line 131: 93mg -> 93 mg. only % and ºC are right after the corresponding values.”

All values have been replaced (page 2).

  1. “Line 168: 1% 168 antibiotic-antimycotic mixture (Gibco)-> is it penicillin-streptomycin?”

Yes, the antibiotic-antimycotic mixture is penicillin-streptomycin and it has been replaced in the revised manuscript.

  1. “Line 192: “Aza-reversine or its biotinylated form (100 nΜ/well) was added in the cultures the next day” – shouldn’t you wait until cells adapt further to the new substrate and reach confluency (or close to it)? You need to describe the state of your culture at each step of the protocol and the passage number”

All information is given in the revised manuscript, please see page 3.

  1. Scheme 80 should have higher resolution. Why 80?

Scheme has been replaced (page 4).

  1. “Figure 1: Again, you need to show your results with higher-resolution images, and larger images in this case”

Figure 1 has been replaced with a higher-resolution image. Also scale bars were added (page 4).

  1. “You observed morphological changes. But can you say that it induces dedifferentiation? So, which cell phenotype are you observing at the end of this particular protocol? You need further characterization. To attest osteogenic differentiation, you used an early and late-stage bone differentiation marker. Add at least 1-2 dedifferentiation markers. Or are you looking at transdifferentiation instead? The paper with doi: 10.1038/s41598-021-91468-w discusses that possibility. These aspects should be clear from your analyses”

The observed morphological changes have been utilized with optical microscope. That is true. On the other hand within this paper you included below is discussing the reverse of the lineage-committed murine myoblasts to a more primitive multipotent state and their differentiate into osteoblasts and adipocytes under lineage-specific inducing conditions. By taking into account both the observed effect upon the addition of reversine as well as the used terminology we used the reprogramming instead of the reverse to a more primitive multipotent state with end stage the osteogenesis that is shown as briefly described below. In our manuscript we investigated the reprogramming of MRC5 into osteocytes and that is justified and indicated not only enzymicaly (alkaline phosphatase) or with colorimetric methods (Alizarin red) but also following the expression of osteocalcin gene, a marker of osteogenesis, by using molecular biology methodologies.

  1. “ALP can also be indicative of stemness. Can you elaborate on this?”

According to Štefková et al., (2015) ALP cannot be used as a universal marker for stem cells and high levels of AP is associated with the process of differentiation rather than with stemness. Also, in studies conducted is used as an early-stage marker of stemness while in osteogenic differentiation its highest expression is noted at the 14th day of differentiation.

  1. “I am also troubled with your control. Why do you use DMSO? Why not simply add a condition without your aza-reversine 2?”

DMSO was used as a control because aza-reversine and its derivate where dialyzed in <1% DMSO. It has been added in the revised manuscript

  1. “Figure 2h: your housekeeping gene is highly variable. This should not happen. Please validate your data with a suitable housekeeping gene”

As represented in figure 2B (A) the housekeeping gene (Gus B) has the same intensity. Figure 2B (b) represents the rt-PCR results for the expression of osteocalcin. Also, quantification of band intensity with GelAnalyser has been added in the supplementary data.

  1. “Figure 4: indeed, you do provide hints that osteogenic differentiation of (apparently) MSCs is enhanced if the medium is supplemented with your biomolecules. Unfortunately, your characterization is somewhat limited. You could add for instance quantification of the alizarin red staining. I know that at least some kits enable qualitative measures but also quantitative one, which would be very interesting in your case. Well, similarly as you did for Figure 7.”

Indeed it could be very interesting to include quantitative measurements, but the basic aim of the study was to investigate if immobilized biotinylated aza-reversine is capable of reprogramming MRC-5 cells or differentiating stem cells and that is the reason we only represent qualitative data for this experiment. Also, both diagrams at figure 7 include quantification experiments for free aza-reversine (page 7)

  1. “Conclusions: this section is too vague and do not summarize all results obtained, highlighting the most important conclusions deriving from those results.”

Conclusion has been rewritten in order to summarize the most important findings of the present study.

Reviewer 2 Report

Dear authors,

As suggested by its title, the article entitled “Aza-reversine Promotes Reprogramming of Lung (MRC-5) and 2 Differentiation of Mesenchymal Cells into Osteoblasts” evaluates MSC and MRC-5 morphological changes following incubation with aza-reversine derivatives and subsequent osteogenic differentiation. Content doesn’t seem to be new, based on the content of the published papers with doi: 10.1038/s41598-021-91468-w, 10.1007/s12291-018-0800-8, 10.1371/journal.pone.0158587, 10.7150/ijbs.12199 as well as paper with ISSN 0393974X. Maybe some novelty exists based on the derivatives produced, but no comparison to reversine is properly done. If you do that, if you prove that your derivatives are new, and are better than classic reversine, that is OK by me. Your outputs are also very limited, and even those are presented in a poor manner, with low resolution images. You do provide interesting hints, but you do need to present a more robust characterization. Intended application could also be more clear from the written text. We cannot foresee what you will do next, or what can be done.

Abstract

“Reversine was originally identified as a small molecule that induces dedifferentiation to C2C12 myoblasts, which were redirected into lipocytes or osteoblasts under lineage specific conditions (LISCs)” – dedifferentiation to C2C12 from which cells? For which clinical needs would that be relevant for? Then you redirect them into lipocytes or osteoblasts? Were you trying to show differentiation potential, or something else? Indeed, this first part of your abstract is a bit unclear. Please justify these aspects a little further.

“After aza-reversine 2 synthesis” – after synthesis of a piperazine analogue of reversine, also known as aza-reversine 2

Could you then focus on a particular application, such as cancer treatment, as you indicate in the Introduction section?

Materials and Methods

Line 131: 93mg -> 93 mg. only % and ºC are right after the corresponding values.

Line 168: 1% 168 antibiotic-antimycotic mixture (Gibco)-> is it penicillin-streptomycin?

Line 192: “Aza-reversine or its biotinylated form (100 nΜ/well) was added in the cultures the next day” – shouldn’t you wait until cells adapt further to the new substrate and reach confluency (or close to it)? You need to describe the state of your culture at each step of the protocol and the passage number. Moreover, for MSCs, you need first to demonstrate that they are indeed MSCs (according to Dominici and colleagues, doi: 10.1080/14653240600855905).

Results:

Scheme 80 should have higher resolution. Why 80?

Figure 1: the scale is missing. Again, you need to show your results with higher-resolution images, and larger images in this case.

You observed morphological changes. But can you say that it induces dedifferentiation? So, which cell phenotype are you observing at the end of this particular protocol? You need further characterization. To attest osteogenic differentiation, you used an early and late-stage bone differentiation marker. Add at least 1-2 dedifferentiation markers. Or are you looking at transdifferentiation instead? The paper with doi: 10.1038/s41598-021-91468-w discusses that possibility. These aspects should be clear from your analyses.

ALP can also be indicative of stemness. Can you elaborate on this?

I am also troubled with your control. Why do you use DMSO? Why not simply add a condition without your aza-reversine 2?

Figure 2h: your housekeeping gene is highly variable. This should not happen. Please validate your data with a suitable housekeeping gene.

Figure 4: indeed, you do provide hints that osteogenic differentiation of (apparently) MSCs is enhanced if the medium is supplemented with your biomolecules. Unfortunately, your characterization is somewhat limited. You could add for instance quantification of the alizarin red staining. I know that at least some kits enable qualitative measures but also quantitative one, which would be very interesting in your case. Well, similarly as you did for Figure 7.

Conclusions: this section is too vague and do not summarize all results obtained, highlighting the most important conclusions deriving from those results.

Author Response

(The authors gave the same response as above.)

Round 2

Reviewer 1 Report

The authors have addressed my comments. I support publication

Reviewer 2 Report

Thank you for your effort in responding to my doubts and concerns. The quality of the paper has been improved.